# Race and Ethnic Differences in the Protective Effect of Parental Educational Attainment on Subsequent Perceived Tobacco Norms among US Youth

**DOI:** 10.3390/ijerph20032517

**Published:** 2023-01-31

**Authors:** Edward Adinkrah, Babak Najand, Angela Young-Brinn

**Affiliations:** 1Department of Family Medicine, Charles R Drew University of Medicine and Science, Los Angeles, CA 90059, USA; 2Marginalization-Related Diminished Returns, Los Angeles, CA 90059, USA

**Keywords:** population groups, risk behavior, perceived tobacco use norms, ethnic groups, academic achievement

## Abstract

Background: Although parental educational attainment is known to be associated with a lower prevalence of behaviors such as tobacco use, these effects are shown to be weaker for Black than White youth. It is important to study whether this difference is due to higher perceived tobacco use norms for Black youth. Aim: To study the association between parental educational attainment and perceived tobacco use norms overall and by race/ethnicity among youth in the US. Methods: The current study used four years of follow-up data from the Population Assessment of Tobacco and Health (PATH-Youth) study conducted between 2013 and 2017. All participants were 12- to 17-year-old non-smokers at baseline and were successfully followed for four years (*n* = 4329). The outcome of interest was perceived tobacco use norms risk at year four. The predictor of interest was baseline parental educational attainment, the moderator was race/ethnicity, and the covariates were age, sex, and parental marital status at baseline. Results: Our linear regressions in the pooled sample showed that higher parental educational attainment at baseline was predictive of perceived disapproval of tobacco use at year four; however, this association was weaker for Latino than non-Latino youth. Our stratified models also showed that higher parental educational attainment was associated with perceived tobacco use norms for non-Latino but not for Latino youth. Conclusion: The effect of high parental educational attainment on anti-tobacco norms differs between Latino and non-Latino youth. Latino youth with highly educated parents remain at risk of tobacco use, while non-Latino youth with highly educated parents show low susceptibility to tobacco use.

## 1. Introduction

Youth is associated with heightened risk behaviors, including tobacco use [1]. However, socioeconomic status (SES) indicators such as parental educational attainment may lower youth risk-taking behaviors such as tobacco use [2]. Some of the many mechanisms that may explain the lower behavioral and health risk of high SES youth are social norms and beliefs that are not favorable toward tobacco (also called perceived tobacco use norms) [3], which are under the influence of peers, families [4], and other factors such as availability of tobacco in the areas, tobacco ads, and prevalence of tobacco use in the community, neighborhood, school, and family and friends [5].

However, the protective effects of parental educational attainment on youth risk behaviors such as tobacco use may differ between diverse racial and ethnic groups of youth [6]. In addition, according to a phenomenon called marginalization-related diminished returns (MDRs) [7,8,9,10,11,12,13,14,15,16], due to racism and social stratification, resources and assets may be associated with lower levels of economic, behavioral, developmental, and health outcomes for marginalized and racialized groups than White individuals [17,18].

Research has indicated that race may alter how SES influences health and behavioral problems such as tobacco use [19,20,21,22,23,24,25,26,27,28,29]. The association between parental educational attainment and a wide range of health problems varies between racial/ethnic groups of youth [30,31,32]. Fuller Rowell showed that the association between youth educational attainment and health is racialized [30,31,32]. Under racism and discrimination, high educational attainment may be linked to more distress and discrimination for Black than White youth [30,31,32]. Education gains may be linked to worse mental health for Black youth who live in a social context that may impose a higher level of psychological tax for their educational success or chronic poverty from childhood [30,31,32]. At all SES levels, Black students are discriminated against [33,34], and high SES Black youth attend worse schools than White youth [35]. Similarly, high-SES Black youth have family members who are more likely to be substance users than high-SES White youth [36]. When high-SES Black youth move to high-SES neighborhoods and schools (that are predominantly White), they become even more exposed [37,38] and vulnerable [39] to discrimination. As the education system differently treats Black and White youth [40,41], health gain due to education is weaker for Blacks than Whites [30,31,32]. 

According to the marginalization-related diminished returns (MDRs), SES resources and even non-economic resources may generate fewer behavioral, developmental, and health outcomes for marginalized and racialized groups such as Blacks and Latinos than non-Latino Whites [17,18]. While most of this literature is generated on the effects of SES on health outcomes for adults [16,19,21,23,29,42,43,44], non-SES factors such as self-efficacy may also be associated with lower health gain for Black than White individuals [45]. Similarly, positive affect [46,47] and happiness [48,49,50] may generate less health for Blacks than Whites. We explain this phenomenon through racism and societal inequalities: Even when SES and other resources are available, societal and environmental conditions such as social stratification, segregation, racism, and discrimination make it more difficult for Black and Latino than non-Latino White families and individuals to secure outcomes. In this view, what makes a large change for Whites may generate smaller real-life changes for Black individuals [45,51].

As shown by systematic reviews, behaviors such as tobacco consumption are under influence of cognitive elements such as perceived tobacco norms [52]. According to theories such as Theory of Planned Behavior (TPB) [52] and Theory of Reasoned Action (TRA) [53], perceived norms predict behaviors such as tobacco use. Perceived norms are different than actual norms and can be defined as what individuals think are the norms of their group [54]. For example, even when actual norms can be low, perceived norms can be high. Thus perceived norms are what people think is the norm, while actual norm is the reality of the society [55]. Cognitive elements such as perceived tobacco norms can be used as a marker of tobacco susceptibility and vulnerability [56].

Built on the MDRs literature on tobacco use risk [57,58], we conducted this study with two aims: the first was to test the association between parental educational attainment and perceived tobacco use norms overall. The second aim was to test the variation of this association by race. Our first hypothesis was that overall, high parental educational attainment is associated with lower perceived tobacco use norms in youth. Our second hypothesis was that this inverse association would be weaker for Latinos and Blacks than non-Latinos and Whites. 

## 2. Methods

For this study, we conducted a secondary analysis of the first four years of the Population Assessment of Tobacco and Health (PATH-Youth) study data. The PATH-Youth is the state-of-the-art study of tobacco use of US youth. Data collection was performed between 2013 (baseline) and 2017 (follow up). Youth PATH data are publicly available to all individuals. This data set is fully de-identified and can be accessed here: https://www.icpsr.umich.edu/web/NAHDAP/studies/36231 (accessed on 12 October 2022).

In the PATH study, participants are selected randomly. Stratified and clustered random samples were selected from all US states. Eligibility for inclusion in the current analysis were non-institutionalized members of US households, aged between 12 and 17 at baseline, having follow-up data for years (baseline and follow-up data), and being Latino or non-Latino White or Black. Participants were all never smoker at baseline. A total number of 4596 youth were entered who had and follow-up data for four years.

Study variables in this analysis included race, ethnicity, parental educational attainment, parental marital status, age, sex/gender, and perceived tobacco use norms. Age was a dichotomous variable 0 for lower than 15 and 1 for 15 and above. Gender was 1 for males and 0 for females. Parental educational attainment was the independent variable with five levels, and perceived tobacco use norms were the outcome. Both parental educational attainment and perceived tobacco use norms were treated as continuous measures. Perceived tobacco use norms were self-reported and measured using the following binary indicators: (a) People who are important to you: Their views on tobacco use in general, (b) People who are important to you: Their views on smoking cigarettes, (c) People who are important to you: Their views on using e-cigarettes or other electronic nicotine products, (d) People who are important to you: Their views on smoking traditional cigars, cigarillos, or filtered cigars, (e) People who are important to you: Their views on smoking shisha or hookah tobacco, (f) People who are important to you: Their views on using snus, and (g) People who are important to you: Their views on other types of smokeless tobacco. Each item was on a 1 (very positive) to 5 (very negative) response scale. The range of total scores was between 1 and 5, with a higher score indicating higher perceived tobacco use norms.

*Parental educational attainment.* Parental educational attainment was a five-level variable as below: 1 = “Some high school,” 2 = “Completed high school,” 3 = “Some college,” 4 = “Completed college,” 5 = “Graduate or professional school after college.” This variable was a continuous variable.

*Parental marital status.* Parental marital status was a dichotomous variable that reflected married parents and any other condition (divorced, not married, partnered, etc.).

*Race.* Race was self-identified, treated as a nominal variable, and the moderator variable (White and Black). Race was the effect modifier (moderator). In this study race was a social rather than a biological variable. White was defined as a person having origins in any of the original peoples of Europe, the Middle East, or North Africa. Black or African American was defined as a person having origins in any of the Black racial groups of Africa. We used race as an effect modifier because MDRs theory suggests that due to racism and social stratification, returns of SES indicators such as parental education tend to be weaker for racialized groups.

*Ethnicity.* Ethnicity was self-identified as non-Latino, or Latino. We defined Latino as “a person of Cuban, Mexican, Puerto Rican, South or Central American, or other Spanish culture or origin regardless of race”.

### Data Analysis

Data analysis was performed using SPSS 24. SPSS was used for univariate, bivariate, and multivariable analysis. Univariate was descriptive statistics such as mean (standard deviation [SD]) and frequency (%). Bivariate included the Spearman correlation test. With the outcome being perceived tobacco use norms score at age 4, the predictor variable was parental educational attainment, and the moderators (effect modifiers) were race and ethnicity, and age, sex, and parental marital status as the covariates, six linear regression models were applied for multivariable modeling. *Model 1* and *Model 2* were run in the pooled sample. *Model 3* and *Model 4* were performed on non-Latino and Latino youth. *Model 5* and *Model 6* were performed on White and Black youth. *Model 1* did not have, and *Model 2* had the interaction term between race/ethnicity and parental education, our predictor variable. *Model 5* and *Model 6* were not shown because there were no race differences in associations. *Model 7* to *Model 10* were performed in race × ethnic groups. *Model 11* and *12* were performed by sex/gender. B, SE, 95% CI, and *p* were reported from each model.

## 3. Results

### 3.1. Descriptive Data 

A total number of 4815 youth were entered who had and follow-up data for four years. Descriptive data are reported in Table 1. 

### 3.2. Pooled Sample Models

Table 2 presents the summary of linear regressions for *Model 1* and *Model 2* that were fitted to the pooled sample. As this model shows, higher parental educational attainment was associated with lower perceived tobacco use norms; however, this association was stronger for non-Latino than Latino youth. White and Black youth did not show difference in the slope of the effect of parental educational attainment on outcome.

### 3.3. Ethnic Stratified Models

Table 3 presents the summary of linear regressions for *Model 3* and *Model 4* that were fitted to White and Black youth, respectively. As these models show, higher parental educational attainment was associated with a lower perceived tobacco use norms for non-Latino but not for Latino youth.

### 3.4. Race × Ethnic Interactional Stratified Models

As shown by *Models 5 to 8* performed in race by ethnic intersectional groups, parental education was associated with higher perceived tobacco use norms score in non-Latino Whites and non-Latino Blacks. This association was not significant for Latino White and Latino Black individuals (Table 4).

### 3.5. Sex/Gender Stratified Models

Due to low sample size, interaction between race or ethnicity with parental education did not show significance in our male or female youth. Table 5 shows the summary of these findings.

## 4. Discussion

The current study was performed with two main aims: one to evaluate the overall association between parental educational attainment and perceived tobacco use norms in US youth, and two to test variation in this association by race and ethnicity. The first aim showed an inverse association between parental educational attainment and perceived tobacco use norms overall. The second aim showed moderation by ethnicity not race. This protective association was weaker for Latino than non-Latino youth. This association did not differ between Black and White youth.

The inverse association between parental educational attainment and perceived tobacco use norms is in line with theories of fundamental causes, social determinants, social status, status syndrome, and several other models that explain the lower risk of high SES populations and individuals. Due to Jim Crow, historical racism, the legacy of slavery, social stratification, and segregation, Black-White differences in living conditions sustain across all levels of socioeconomic inequalities [59,60,61,62]. According to ecological theories, individuals who live in proximity to low SES neighborhoods, peers, schools, families, and friends will have a higher risk, including tobacco use risk [63]. However, many mechanisms may explain why low SES is associated with race, peer risk, and poor neighborhoods.

There are multiple studies that show racial and ethnic variation in the association between SES, health, and behaviors, with weaker associations in racial and ethnic minorities than non-Latino White youth [64]. There are also studies showing weaker associations between SES and tobacco risk in Black and Latino than non-Latino White individuals [16,19,20,21,22,23,25,26,27,28,29,65]. However, we are unaware of any past studies on racial and ethnic differences in the association between parental educational attainment and perceived tobacco use norms.

There are several studies on racial and ethnic variation in health-behavior association [30,31,32]. One of their studies showed that Black and Native American adolescents pay greater social costs with academic success than Whites; however, this is seen in high-achieving schools with a smaller percentage of Black students [32]. In another study, they showed that the effects of educational attainment were weaker for Black than for whites, and only 8% of this difference was due to covariates. Analyses yielded consistent results. They concluded that the effects of educational attainment on inflammation levels are stronger for whites than for racial and ethnic minorities [31].

Most past research is conducted on Black, not Latino individuals. Our observation of a weaker association between parental educational attainment and perceived tobacco use norms in Latino than non-Latino youth is also in line with many previous publications on the MDRs. According to marginalization-related diminished returns, resources and assets generate fewer economic, behavioral, developmental, and health outcomes for marginalized groups than for White individuals. While most of this literature is generated on SES effects among adults, there are some studies showing that a sense of mastery, agency, and self-efficacy may be associated with lower health for Black than White individuals [45]. Similarly, positive affect [46,47], happiness [48,49,50], and a sense of health [66,67,68] may generate more life expectancy for Whites than Blacks [45,51]. The positive association between SES and John Henryism is also suggestive of the health risks that may be the price of success for Black individuals [69,70,71,72,73]. Hudson has published on the high costs of success for Black youth and young adults [70,74,75].

This study expanded the MDRs literature, which is written on tobacco use [57,58]. Previous work has shown that SES –tobacco use is racialized [57,58]. A study showed that education–tobacco knowledge is also racialized in the US [76]. This finding may be because high-SES White youth attend better schools than high-SES Black youth [35]. In addition, there are many challenges in the daily lives of Black youth in US schools [33,34]. Racial differences in the returns of education may be because of anti-Black discrimination at schools [33,34] or neighborhoods [37,38].

Our study is not without methodological limitations. First, all variables were self-report. Thus, our results may be affected by reporting bias and social desirability. Second, our variables were measured from youth. Norms could be measured from the social network of the youth. We did not measure many potential confounders, such as drug availability at home or neighborhood conditions, such as proximity to tobacco outlets. In addition, this was a study with an imbalanced sample size (larger n for non-Latino and White than Latino and Black youth). However, our main inference was based on pooled sample analysis with interaction rather than stratified models, which have differential power. Our study explored sex differences in the relationship between parental educational attainment and youth’s perceived tobacco use norms, however, the sample size was inadequate for race by sex by parental education interaction term. Despite these limitations, the major contribution of this study is to document MDRs for perceived tobacco use norms for the first time. We are not aware of any previous studies that suggest perceived tobacco use norms may have a role in higher-than-expected tobacco use of Black and Latino youth with highly educated parents.

Future research is needed on the social and environmental causes of the observed MDRs. Future research should test the role of advertisement exposure, the prevalence of smokers, as well as other contextual factors at school and neighborhood that may weaken the effect of parental educational attainment for ethnic minority youth. The role of high-risk peers, family, friends, proximity to tobacco outlets, and other contextual conditions should be tested in future multi-level research.

## 5. Conclusions

To conclude, although overall, high parental educational attainment is associated with lower perceived tobacco use norms, this inverse association is weaker for Latino than non-Latino youth. The diminished return of parental educational attainment on perceived tobacco use norms may be due to environmental and structural inequalities at family, school, or neighborhood due to the segregation of ethnic minority communities. Future research should test why and how the same MDRs could not be found for Black youth.

## Figures and Tables

**Table 1 ijerph-20-02517-t001:** Descriptive data overall and by race in youth (*n* = 4329).

	All		Non-Latino White		Non-Latino Black		Latino White		Latino Black	
	*n* 4596	%	*n* 2507	%	*n* 757	%	*n* 966	%	*n* 99	%
12–14	4433	96.5	2427	96.8	714	94.3	938	97.1	96	97.0
15–18	163	3.5	80	3.2	43	5.7	28	2.9	3	3.0
Sex/Gender										
Female	2199	47.8	1201	47.9	367	48.5	455	47.1	49	49.5
Male	2384	51.9	1303	52.0	385	50.9	510	52.8	47	47.5
Marital Status of the Parents										
Not Married	1653	36.0	665	26.5	470	62.1	364	37.7	69	69.7
Married	2943	64.0	1842	73.5	287	37.9	602	62.3	30	30.3
Parental educational attainment (1–5)	2.7963	1.25674	3.2110	1.17118	2.6222	1.16380	2.1460	1.14105	2.2828	1.16969
Perceived tobacco use norms (1–5)	4.2577	0.80773	4.2525	0.81754	4.1477	0.87378	4.3365	0.73895	4.2225	0.83277

**Table 2 ijerph-20-02517-t002:** Pooled Sample models in US youth.

	Unstandardized B	Unstandardized Std. Error	Standardized Beta	Lower Bound	Upper Bound	Sig.
**Model 1 (All, Main Effects)**						
Race (Black)	−0.071	0.032	−0.036	−0.134	−0.009	0.025
Ethnicity (Latino)	0.175	0.031	0.095	0.115	0.236	0.000
Male	0.039	0.025	0.024	−0.010	0.087	0.117
Age	−0.029	0.067	−0.007	−0.160	0.102	0.664
Parental Educational Attainment (1–5)	0.106	0.011	0.165	0.085	0.126	0.000
**Model 2 (All, M1 + Race Interaction)**						
Race (Black)	−0.014	0.078	−0.007	−0.166	0.138	0.857
Ethnicity (Latino)	0.179	0.031	0.097	0.118	0.240	0.000
Male	0.039	0.025	0.024	−0.010	0.087	0.118
Age	−0.030	0.067	−0.007	−0.161	0.101	0.653
Parental Educational Attainment (1–5)	0.110	0.012	0.171	0.087	0.133	0.000
Parental educational attainment (1–5) × Race (Black)	−0.021	0.026	−0.031	−0.073	0.030	0.418
**Model 2 (All, M1 + Ethnicity Interaction)**						
Race(Black)	−0.053	0.032	−0.026	−0.115	0.010	0.099
Male	0.486	0.067	0.263	0.354	0.618	0.000
Age	0.037	0.025	0.023	−0.012	0.085	0.136
Married Parents	−0.029	0.067	−0.007	−0.160	0.102	0.663
Parental educational attainment (1–5)	0.137	0.012	0.214	0.113	.161	0.000
Parental educational attainment (1–5) × Ethnicity (Latino)	−0.129	0.025	−0.176	−0.178	−0.080	0.000

Outcome: Perceived tobacco use norms Score; Data: Population Assessment of Tobacco and Health (PATH).

**Table 3 ijerph-20-02517-t003:** Stratified models in non-Latino and Latino youth.

	Unstandardized B	Unstandardized Std. Error	Standardized Beta	Lower Bound	Upper Bound	Sig.
**Model 3 (Non-Latino)**						
Race (Black)	−0.042	0.035	−0.022	−0.111	0.026	0.228
Male	0.018	0.029	0.011	−0.039	0.075	0.538
Age	−0.044	0.076	−0.010	−0.193	0.104	0.556
Parental educational attainment (1–5)	0.138	0.012	0.201	0.113	0.162	0.000
**Model 4 (Latino)**						
Race (Black)	−0.114	0.081	−0.044	−0.273	0.044	0.158
Male	0.094	0.046	0.063	0.003	0.185	0.044
Age	0.041	0.144	0.009	−0.242	0.324	0.776
Parental educational attainment (1–5)	0.008	0.020	0.013	−0.031	0.048	0.677

Outcome: Perceived tobacco use norms Score; Data: Population Assessment of Tobacco and Health (PATH).

**Table 4 ijerph-20-02517-t004:** Models in race × ethnicity groups.

	Unstandardized B	Unstandardized Std. Error	Standardized Beta	Lower Bound	Upper Bound	Sig.
**Model 5 (Non-Latino White)**						
Age	−0.070	0.090	−0.015	−0.247	0.107	0.438
Male	0.023	0.032	0.014	−0.040	0.085	0.475
Parental educational	0.164	0.014	0.236	0.138	0.191	0.000
**Model 6 (Non-Latino Black)**						
Age	0.025	0.137	0.007	−0.244	0.293	0.856
Male	0.047	0.063	0.027	−0.077	0.172	0.456
Parental educational	0.093	0.027	0.124	0.040	0.147	0.001
**Model 7 (Latino White)**						
Age	0.100	0.141	0.023	−0.178	0.377	0.481
Male	0.132	0.048	0.089	0.039	0.225	0.006
Parental educational	0.005	0.021	0.008	−0.036	0.046	0.805
**Model 8 (Latino Black)**						
Age	−0.387	0.464	−0.085	−1.308	0.535	0.407
Male	−0.403	0.162	−0.255	−0.724	−0.082	0.015
Parental educational	0.030	0.067	0.045	−0.104	0.164	0.655

Outcome: Perceived tobacco use norms Score; Data: Population Assessment of Tobacco and Health (PATH).

**Table 5 ijerph-20-02517-t005:** Stratified models in male and female youth.

	Unstandardized B	Unstandardized Std. Error	Standardized Beta	Lower Bound	Upper Bound	Unstandardized B	Sig.	Unstandardized Std. Error	Standardized Beta	Lower Bound	Upper Bound	Sig.
**Females**												
Race (Black)	−0.022	0.044	−0.011	−0.109	0.064	0.120	0.610	0.106	0.060	−0.088	0.328	0.257
Ethnicity (Hispanic)	0.181	0.040	0.102	0.102	0.260	0.186	0.000	0.044	0.099	0.100	0.271	0.000
Age	−0.125	0.092	−0.029	−0.305	0.055	−0.099	0.172	0.095	−0.023	−0.286	0.087	0.297
Parent eucation	0.114	0.014	0.178	0.086	0.142	0.133	0.000	0.017	0.206	0.101	0.166	0.000
Parent eucation × Race (Black)						−0.052		0.036	−0.076	−0.123	0.018	0.147
**Males**												
Race (Black)	−0.078	0.044	−0.037	−0.164	0.007	−0.053	0.073	0.110	−0.026	−0.269	0.163	0.630
Ethnicity (Hispanic)	0.213	0.040	0.119	0.135	0.291	0.215	0.000	0.043	0.113	0.131	0.299	0.000
Age	0.053	0.088	0.012	−0.119	0.225	0.062	0.547	0.091	0.014	−0.116	0.240	0.496
Parent eucation	0.108	0.014	0.167	0.080	0.135	0.109	0.000	0.016	0.166	0.077	0.140	0.000
Parent eucation × Race (Black)						−0.009		0.038	−0.012	−0.083	0.065	0.811

Outcome: Perceived tobacco use norms Score; Data: Population Assessment of Tobacco and Health (PATH).

## Data Availability

PATH data are publicly available here: https://www.icpsr.umich.edu/web/NAHDAP/series/606 (accessed 12 October 2022).

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
