# Peer review of "Race and Ethnic Differences in the Protective Effect of Parental Educational Attainment on Subsequent Perceived Tobacco Norms among US Youth"

_ijerph, 2023, doi:10.3390/ijerph20032517_

Round 1

Reviewer 1 Report

1.      As this study investigate race and ethnic differences in the protective effect of parental education attainment on subsequent perceived tobacco norms among US youth it would be useful to know looking at table one which race are other participants. Authors mentioned black, white and other. Due to the aim of the study I think they should provide detail answer.

2.      Also from the table one we can see that ethnicity is non-latino and latino. The race black, white and other altogether sum 4815 participants. Latino and non-latino participants also sum 4815 participants. What does it mean latino ethnicity? Participants with latino ethnicity belong to which race? And also with non – latino ethnicity? Please provide data for all participants which race and ethnicity they are belonging to. It should be discussed in more detail.

3.      It would be useful to discuss the effect of parental education attainment on subsequent perceived tobacco norms in every subgroup. For example black latino and white latino, and black non-latino and white non-latino. Are there any notable differences?

4.      In the methods authors mentioned that both parental educational attainment and perceived tobacco use norms were treated as continuous measures (lines 96-97). There is no need to repeat that in lines 110-111.

5.      It must be explained why race is effect modifier, and how that affected study results.

6.      In the study limitations authors mentioned that they did not evaluate sex differences in the relationship between parental educational attainment and youth's perceived tobacco use norms (lines 211-213). Please, provide why the analysis was not performed.

7.      Also in the study limitations was mentioned that this was a study with an imbalanced sample size (larger n for non-Latno and White than Latino and Black youth). It must be discussed in more detail because according to data there are overlapping between data and race, because there are non-latino and black participants and latino and white, isn’t it?

8.      Why age was analyzed as dichotomus variable of 0 for lower than 15 and 1 for 15 and above?

9.      If the study used publicly available PATH data and all data are fully de-identified how that is in the line with longitudinal study?  Who was observed and followed for four years and how when data are fully de-identified?

1.  In table one is written feale instead of female, in table two model 2 race interaction the last row is relocated.

Author Response

COMMENT

RESPONSE

As this study investigate race and ethnic differences in the protective effect of parental education attainment on subsequent perceived tobacco norms among US youth it would be useful to know looking at table one which race are other participants. Authors mentioned black, white and other. Due to the aim of the study I think they should provide detail answer.

White – A person having origins in any of the original peoples of Europe, the Middle East, or North Africa.

Black or African American – A person having origins in any of the Black racial groups of Africa.

OTHER:

American Indian or Alaska Native – A person having origins in any of the original peoples of North and South America (including Central America) and who maintains tribal affiliation or community attachment.

Asian – A person having origins in any of the original peoples of the Far East, Southeast Asia, or the Indian subcontinent including, for example, Cambodia, China, India, Japan, Korea, Malaysia, Pakistan, the Philippine Islands, Thailand, and Vietnam.

Native Hawaiian or Other Pacific Islander – A person having origins in any of the original peoples of Hawaii, Guam, Samoa, or other Pacific Islands.

Also from the table one we can see that ethnicity is non-latino and latino. The race black, white and other altogether sum 4815 participants. Latino and non-latino participants also sum 4815 participants. What does it mean latino ethnicity? Participants with latino ethnicity belong to which race? And also with non – latino ethnicity? Please provide data for all participants which race and ethnicity they are belonging to. It should be discussed in more detail.

According to the U.S. Office of Management and Budget (OMB), and these data are based on self-identification.

For ethnicity, the OMB standards classify individuals in one of two categories: “Hispanic or Latino” or “Not Hispanic or Latino.” We use the term “Hispanic or Latino” interchangeably with the term “Hispanic,” and also refer to this concept as “ethnicity.”

The OMB standards also emphasize that people of Hispanic origin may be of any race. In data tables, we often cross-tabulate the race and Hispanic origin categories to display Hispanic as a single category and the non-Hispanic race groups as categories summing up to the total population.

Also, according to the NIH;

Latino/a or Latinx

A person whose origins are in Latin America, including Cuba, Mexico, Puerto Rico, South America, or Central America. Latino is reserved for men and Latina for women. The plural Latinas is for a group of women and Latinos is for a group of men. A mixed gender group of Latin American descent, however, would revert to the masculine Latinos.

Hispanic

A person descended from Spanish-speaking populations. People who identify their origin as Hispanic, Latino, or Spanish may be of any race. Most people with origins in Brazil are considered Latino but not Hispanic because most Brazilians speak Portuguese. Similarly, Spanish people may be considered Hispanic but not Latino. Because the terms are vague, use the more specific geographic origin (Colombian, Honduran, Brazilian), if possible.

It would be useful to discuss the effect of parental education attainment on subsequent perceived tobacco norms in every subgroup. For example black latino and white latino, and black non-latino and white non-latino. Are there any notable differences?

We have very small

In the methods authors mentioned that both parental educational attainment and perceived tobacco use norms were treated as continuous measures (lines 96-97). There is no need to repeat that in lines 110-111.

Corrected/ Deleted.

It must be explained why race is effect modifier, and how that affected study results.

As added to the paper, according to MDRs (see introduction) in addition to its main effect, it alters correlates (gain) of SES such as parental education. The whole MDRs theory conseptualizes race as a moderator. This is clearly explained now.

In the study limitations authors mentioned that they did not evaluate sex differences in the relationship between parental educational attainment and youth's perceived tobacco use norms (lines 211-213). Please, provide why the analysis was not performed.

We now have added sex differences as our new models. Given the low sample size, sex specific models do not show any MDRs, which is due to low power. We added this new results because the other reviewer has also asked about it and you are also interested to observe additional analyses such as sex differences.

Also in the study limitations was mentioned that this was a study with an imbalanced sample size (larger n for non-Latno and White than Latino and Black youth). It must be discussed in more detail because according to data there are overlapping between data and race, because there are non-latino and black participants and latino and white, isn’t it?

Race refers to White vs Black and Ethnicity refers to Latin and not Latino. Here we have used two variables as proxies of marginalizarion. There is no any individual in this study who is Asian, Native American, or other racial groups. This is made clear in the methods section.

Why age was analyzed as dichotomus variable of 0 for lower than 15 and 1 for 15 and above?

The ABCD data do not provide exact age of participant to protect them. So a part of deidentification is to combined the aged this way. So they collectively group youth to younger (below 15) and older (above 15) youth.

If the study used publicly available PATH data and all data are fully de-identified how that is in the line with longitudinal study?  Who was observed and followed for four years and how when data are fully de-identified?

Retrospective review of PATH data.

Original dataset from PATH already de-identified

In table one is written feale instead of female, in table two model 2 race interaction the last row is relocated.

Both corrected

Reviewer 2 Report

The summary should mention the methodology used, the sample size at the beginning and the number of participants, and should explain more about the conclusions and less about the limitations.

Introduction. It is not explained why it is necessary to carry out this type of research and above all it is not explained why it is being carried out.

In terms of background, there are no theories or previous research that explain the behaviour of tobacco consumption and which are so necessary to understand such a complex behaviour as tobacco consumption. 

Subjects and methods.

The procedure for obtaining the PATH-Youth sample is not explained.

They should explain more about the contributions of this study.  The authors themselves recognise that this study has many limitations.

Inclusion and exclusion criteria are not explained.

The period and date of data collection are not specified.

Data should also be stratified by sex.

Author Response

Thank you for your excellent comments. They have improved the paper. We have made changes to the paper. These changes are in shown in yellow.

The summary should mention the methodology used, the sample size at the beginning and the number of participants, and should explain more about the conclusions and less about the limitations.

These changes are made to the abstract/summary.

Introduction. It is not explained why it is necessary to carry out this type of research and above all it is not explained why it is being carried out. In terms of background, there are no theories or previous research that explain the behaviour of tobacco consumption and which are so necessary to understand such a complex behaviour as tobacco consumption. 

A paragraph is now added to the paper background that highlights the role of perceived norm as a cognitive determinant of tobacco use.

Subjects and methods.

The procedure for obtaining the PATH-Youth sample is not explained.

The procedure for obtaining the PATH-Youth sample is explained in the methods section.

They should explain more about the contributions of this study.  The authors themselves recognise that this study has many limitations.

We added half a paragraph to the discussion after limitation (same paragraph).

Inclusion and exclusion criteria are not explained.

We added this information to the paper.

The period and date of data collection are not specified.

This information is added to the methods of the paper. 

Data should also be stratified by sex.

The last table reports the results for males and females. As sample size is half, these interactions are not significant in either of the genders, again because of the sample size.

Round 2

Reviewer 1 Report

The uploaded paper has improved a lot.